# OpenReview forum: "Adversarial Tuning: Defending Against Jailbreak Attacks for LLMs"
_ICLR.cc/2025/Conference — ICLR 2025 Conference Withdrawn Submission_

### Official Review · Reviewer_e4S8 · 2024-11-02

**Soundness:** 2
**Presentation:** 1
**Contribution:** 2
**Rating:** 3
**Confidence:** 4

**Summary:**

This paper proposes a two-stage adversarial tuning method that optimizes a safe prompt to defend LLMs against jailbreaking. The first stage
focuses on generating token-level adversarial prompts. The second stage automatically generates out-of-distribution adversarial prompts to generalize the safe prompt to unseen attacks.

**Strengths:**

1. The defense considers unseen attacks (i.e. out-of-distribution adversarial prompts), which is an important and overlooked problem.
1. Experiments demonstrate that the defense can almost eliminate jailbreaking attack success rates on multiple models.

**Weaknesses:**

1. Copy editing issues. This paper exhibits many presentation and format issues, including but not limited to:
  - In the Abstract in the forum, the latex command `\fan{` was not removed.
  - Page 2, the paragraph `Semantic-Level Adversarial Prompt` was not started in a new line.
  - Page 10, the margin of Conclusion was significantly squeezed.
  - Page 26, the watermark is covered by the frame.
  - Page 27, Section D.3 is empty.
2. There are 2 existing works using adversarial training on prompts to defend against jailbreaking, but the differences between the proposed method and them are not well discussed.
- Fight Back Against Jailbreaking via Prompt Adversarial Tuning
- Robust prompt optimization for defending language models against jailbreaking attacks.

3. There are no sufficient experiments that discuss the influence of the method on the natural performance of LLMs. Only experiments on 2 models are presented in the appendix.

4. The main experiment fails to include enough comparisons with baselines. Only experiments on 2 models are compared with baselines, but Figure 7 only includes comparison with no defense, weakening the generalizability of the method on different models.

5. The theoretical analysis appears to be straightforward, and it relies on overly strong assumptions.

**Questions:**

See weaknesses.

---

### Official Review · Reviewer_aQZj · 2024-11-04

**Soundness:** 3
**Presentation:** 3
**Contribution:** 3
**Rating:** 3
**Confidence:** 3

**Summary:**

In this paper, the authors propose adversarial tuning, a two-stage jailbreak defense for LLMs. In the first stage, the authors propose meta-universal prompt learning to efficiently generate token-level jailbreak prompts. In the second stage, the authors propose automatic adversarial prompt learning to construct out-of-distribution training data.

**Strengths:**

1 This paper is easy to follow.

2 The framework overview in Figure 1 makes the pipeline very clear.

3 The experimental section is quite solid.

**Weaknesses:**

1 Minor errors are observed.

- in Line 63, "defe es".
- in Line 1042, the referred figure is broken.

I would suggest authors perform proofreading very carefully.

2 The proposed method requires finetuning the parameters of the models. Therefore, we are unable to validate its effectiveness in closed-source models, which undermines its practicality.

3 The time analysis of the method is needed. As far as I know, the pipeline of the proposed method is not very simple. Therefore, comparison is needed to see how much time it indeed can save.

4 The author aims to include a sufficient amount of content in the main text. However, in my opinion, this results in overly bulky content. In addition, this also results in very narrow gaps between different sections (especially the gap between the related work and conclusion section). In my opinion, the authors place the section transferability of adversarial fine-tuning data in the main text is unnecessary. You can place them in the Appendix section to avoid crowding.

5 It seems that authors generate the adversarial sample offline instead of online. I am not sure whether it will increase the risks of models attacked by adaptive attacks.

**Questions:**

1 Although the authors compare the proposed method with some of the popular baselines in the experimental section. However, from my view, closer baselines should be those that study adversarial tuning in previous studies, for example [1] and [2]. Can the proposed method outperform those techniques on the baseline models?

2 Although the authors evaluate model utility on five benchmarks. However, as far as I know, MT-bench [3] is a popular benchmark that evaluates the capability of LLMs. Can the proposed method perform well in this benchmark?

[1] Fight Back against Jailbreaking via Prompt Adversarial Tuning, in NeurIPS 2024.

[2] HarmBench: A Standardized Evaluation Framework for Automated Red Teaming and Robust Refusal, in ICML 2024.

[3] Judging LLM-as-a-Judge with MT-Bench and Chatbot Arena, in NeurIPS 2023.

---

### Official Review · Reviewer_cge7 · 2024-11-04

**Soundness:** 3
**Presentation:** 3
**Contribution:** 3
**Rating:** 6
**Confidence:** 3

**Summary:**

This paper proposes a two-stage adversarial training method to enhance the defense capabilities of LLMs. The first stage focuses on defending against token-level adversarial prompts, and the second stage targets out-of-distribution adversarial prompts. Extensive experimental results demonstrate that the proposed method significantly improves defense performance against jailbreak attacks. The experiments also reveal a trade-off between model utility and robustness, and an advanced method to balance this trade-off is also proposed.

**Strengths:**

* The method is novel.
* The strategy is sound.
* Comprehensive experiments demonstrate that the method significantly improves the robustness.

**Weaknesses:**

* In addition to generalizing to various jailbreak methods, the most important aspect of training a robust LLM is balancing the trade-off between robustness and model utility. Therefore, it is better to demonstrate the performance of both model utility and robustness simultaneously and compare the proposed method to previous work.

* In Section D.4, the results only show that hybrid adversarial tuning can improve model utility, while the defense performance is overlooked.

* The Llama-2-7B already performs well in defending against many kinds of jailbreak attacks. It seems unconvincing to perform experiments on it. It would be better to show the performance of other models, such as Llama-3.1 and Phi-3. Nevertheless, I understand that the rebuttal time may be tight for adding these experiments.

* Why does the GCG only achieve a 15.38% attack success rate (ASR) when attacking Vicuna-13B? According to recent papers, such as HarmBench, it can achieve over 80% ASR.

* More related work should be discussed and compared, for instance, [1].


[1] Xhonneux, Sophie, et al. "Efficient adversarial training in llms with continuous attacks." arXiv preprint arXiv:2405.15589 (2024).

**Questions:**

See weakness.

---

### Official Review · Reviewer_9XVV · 2024-11-08

**Soundness:** 3
**Presentation:** 3
**Contribution:** 2
**Rating:** 5
**Confidence:** 4

**Summary:**

This paper proposes an innovative approach to enhance the security of Large Language Models (LLMs) through efficient adversarial training. Specifically, it introduces two key techniques aimed at improving the generation of adversarial prompts during the adversarial training process. The first technique involves generating token-level adversarial prompts in a hierarchical manner. Initially, a universal adversarial prompt is created, which serves as a foundation for generating individual adversarial prompts. This hierarchical approach significantly accelerates the generation process of token-level adversarial prompts. The second technique focuses on producing a broader range of out-of-distribution (OOD) adversarial prompts. This diversity is crucial for effective adversarial training, enabling the model to better defend against potential future attacks. The experimental results provided in the paper demonstrate the effectiveness of these methods.

**Strengths:**

- The paper addresses an important and timely issue in the field of machine learning, particularly regarding the vulnerabilities of LLMs to adversarial attacks.

- By targeting both seen and unseen attacks, the proposed method enhances its practical applicability and effectiveness in real-world scenarios.

- Figure 1 is well-crafted and effectively illustrates the concepts discussed, aiding in reader comprehension.

- The authors utilize a diverse set of models, datasets, and various attack/defense methods for their evaluations, ensuring a comprehensive assessment of their approach.

**Weaknesses:**

* The term "continuous" in this context requires clarification. Once adversarial training concludes, the model typically enters a deployment phase without further improvements. If "continuous" refers to iterative training during adversarial training, many prior methods also involve multiple training rounds, which may diminish the novelty of this approach.

* While addressing the speed of adversarial prompt generation is a key goal, the reliance on time-consuming token-level methods raises questions. Why not utilize more efficient generation methods, such as PAIR?

* Although starting with universal adversarial prompts may expedite individual prompt generation, there is concern that this approach could reduce diversity among generated prompts, potentially compromising defense efficacy against a wider array of attacks.

* The paper should elucidate the core insight behind generating diverse OOD adversarial prompts. Allowing models to autonomously generate these prompts poses challenges since models tend to resist such requests.

* The results indicating that the proposed method can reduce the attack success rate (ASR) of unknown attacks to nearly zero warrant scrutiny. An explanation is needed for how such impressive performance is achieved, especially given the vastness and complexity of unknown attack spaces that cannot be fully enumerated in adversarial training phases.

**Questions:**

See above.

---

### Note · Authors · 2024-11-18

**Comment:**

I have read and agree with the venue's withdrawal policy on behalf of my co-authors and me.

**Withdrawal Confirmation:**

I have read and agree with the venue's withdrawal policy on behalf of myself and my co-authors.